# AI system for predicting the deterioration of COVID-19 patients in the emergency department

**Farah E. Shamout**[1,*], **Yiqiu Shen**[2,*], **Nan Wu**[2,*], **Aakash Kaku**[2,*], **Jungkyu Park**[3,8*],
**Taro Makino**[3,2,*], **Stanisław Jastrzębski**[3,4,2], **Jan Witowski**[3,4], **Duo Wang**[5], **Ben Zhang**[5],
**Siddhant Dogra**[3], **Meng Cao**[6], **Narges Razavian**[5,3,2], **David Kudlowitz**[6], **Lea Azour**[3],
**William Moore**[3], **Yvonne W. Lui**[3,4], **Yindalon Aphinyanaphongs**[5],
**Carlos Fernandez-Granda**[2,7], **Krzysztof J. Geras**[3,4,2,✉]

[1]Engineering Division, NYU Abu Dhabi
[2]Center for Data Science, New York University
[3]Department of Radiology, NYU Langone Health
[4]Center for Advanced Imaging Innovation and Research, NYU Langone Health
[5]Department of Population Health, NYU Langone Health
[6]Department of Medicine, NYU Langone Health
[7]Department of Mathematics, Courant Institute, New York University
[8]Vilcek Institute of Graduate Biomedical Sciences, NYU Grossman School of Medicine
[*]Equal contribution
✉k.j.geras@nyu.edu

## Abstract

During the COVID-19 pandemic, rapid and accurate triage of patients at the emergency department is critical to inform decision-making. We propose a data-driven approach for prediction of deterioration risk using a deep neural network that learns from chest X-ray images, and a gradient boosting model that learns from routine clinical variables. Our AI prognosis system, trained using data from 3,661 patients, achieves the AUC of 0.786 (95% CI: 0.742-0.827) when predicting deterioration within 96 hours. Our deep neural network indicates informative areas of chest X-ray images to assist clinicians in interpreting the predictions, and performs comparably to two experienced chest radiologists in a reader study. In summary, our findings demonstrate the potential of the proposed system for assisting front-line physicians in the triage of COVID-19 patients.

## 1 Introduction

In recent months, there has been a surge in patients presenting to the emergency department (ED) with respiratory illnesses associated with SARS CoV-2 infection (COVID-19) [1]. Evaluating the risk of deterioration of these patients to perform triage is crucial for clinical decision-making and resource allocation [2]. Data-driven risk evaluation based on artificial intelligence (AI) could, therefore, play an important role in streamlining ED triage. In this work, we present an AI system that performs an automatic evaluation of deterioration risk, based on chest X-ray imaging, combined with other routinely collected non-imaging clinical variables. The goal is to provide support for critical clinical decision-making involving patients arriving at the ED in need of immediate care [1].

## 2 Methods

We included 5,224 exams (5,617 images) collected from 2,943 patients in the training set and 770 exams (832 images) collected from 718 patients in the test set.[1] As shown in Figure 1, we

---

[1]This study was approved by the Institutional Review Board, with ID# i20-00858.

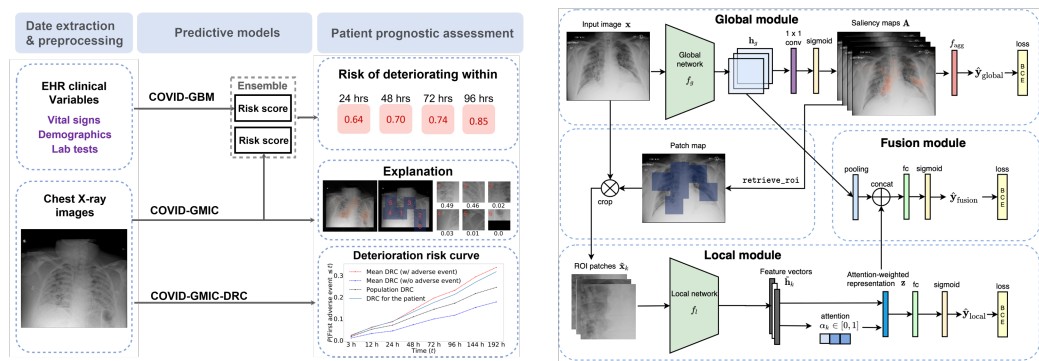

Figure 1: (Left) Overview of the AI system that assesses the patient's risk of deterioration every time a chest X-ray image is collected in the ED. (Right) Architecture of COVID-GMIC.

designed two different models to process the chest X-ray images, both based on the Globally-Aware Multiple Instance Classifier (GMIC) [3]. The first model, COVID-GMIC, predicts the overall risk of deterioration within 24, 48, 72, and 96 hours. COVID-GMIC first processes an X-ray image $\mathbf{x}$ using a global network $f_g$ to generate four saliency maps ($\mathbf{A}^t$) that highlight the regions on the X-ray image that are predictive of the onset of adverse events within $t$ hours, which are then transformed into classification predictions $\hat{\mathbf{y}}_{\text{global}}^t$. COVID-GMIC then applies a local network $f_l$ to extract fine-grained visual details from these regions and computes classification predictions $\hat{\mathbf{y}}_{\text{local}}^t$ based on six regions of interest cropped from the original image. Both $f_g$ and $f_l$ are parameterized as a ResNet-18 [4]. To compute the overall prediction $\hat{\mathbf{y}}_{\text{fusion}}$, COVID-GMIC employs a fusion module that aggregates information from both the global context and local details to make a holistic diagnosis. COVID-GMIC is trained with the following loss function:

$$l(\mathbf{y}, \hat{\mathbf{y}}_{\text{global}}, \hat{\mathbf{y}}_{\text{local}}, \hat{\mathbf{y}}_{\text{fusion}}) = \frac{1}{|\mathbb{T}|} \sum_{t \in \mathbb{T}} \text{BCE}(\mathbf{y}^t, \hat{\mathbf{y}}_{\text{global}}^t) + \text{BCE}(\mathbf{y}^t, \hat{\mathbf{y}}_{\text{local}}^t) + \text{BCE}(\mathbf{y}^t, \hat{\mathbf{y}}_{\text{fusion}}^t) + \beta|\mathbf{A}^t|,$$

where BCE denotes binary cross-entropy, $\beta$ is a hyperparameter representing the weight on an $\ell_1$-norm regularization term that promotes sparsity of the saliency maps and $\mathbb{T} = \{24, 48, 72, 96\}$. Finally, the predictions of COVID-GMIC are combined with predictions of a gradient boosting model [5] that learns from clinical variables, referred to as COVID-GBM, such that $\hat{\mathbf{y}}_{\text{ENSEMBLE}} = \lambda\hat{\mathbf{y}}_{\text{COVID-GMIC}} + (1 - \lambda)\hat{\mathbf{y}}_{\text{COVID-GBM}}$.

The second model, COVID-GMIC-DRC, predicts how the patient's risk of deterioration evolves over time in the form of deterioration risk curves (DRC),

$$l(T, \hat{\mathbf{p}}_{\text{global}}, \hat{\mathbf{p}}_{\text{local}}, \hat{\mathbf{p}}_{\text{fusion}}) = l_s(T, \hat{\mathbf{p}}_{\text{global}}) + l_s(T, \hat{\mathbf{p}}_{\text{local}}) + l_s(T, \hat{\mathbf{p}}_{\text{fusion}}) + \sum_{m=0}^{8} \beta|\mathbf{A}^m|,$$

where $l_s$ is the negative log-likelihood , $\hat{\mathbf{p}} \in \mathbb{R}^8$ and the $i^{th}$ entry of this vector, $\hat{\mathbf{p}}_i$, represents the conditional probability of the adverse event happening before time $t_i \in \{t_i | 1 \leq i \leq 8\}$, where $t_1 = 3, t_2 = 12, t_3 = 24, t_4 = 48, t_5 = 72, t_6 = 96, t_7 = 144, t_8 = 192$ hours. $T$ is the time of the first adverse event.

## 3    Results & Discussion

Our system is able to accurately predict the deterioration risk on a test set of new patients. As shown in Table 1, the ensemble of COVID-GMIC and COVID-GBM, denoted as 'COVID-GMIC + COVID-GBM', achieves the best performance across all time windows in terms of the AUC and precision-recall AUC (PR AUC), except for the PR AUC in the 96 hours task. In particular, it achieves the AUC of 0.786 (95% CI: 0.742-0.827), and the PR AUC of 0.517 (95% CI: 0.434, 0.605) for prediction of deterioration within 96 hours. In the reader study, our main finding is that COVID-GMIC outperforms radiologists A & B across time windows longer than 24 hours, with 3 and 17 years of experience as attending radiologists, respectively. Figure 2 shows the saliency maps produced by COVID-GMIC to illustrate its explainability. There are diffuse airspace opacities, though the saliency maps primarily highlight the medial right basilar and peripheral left basilar opacities. Additionally, the probability of the temporal risk evolution estimated by our system discriminates effectively between patients, and is well-calibrated as shown in Figure 3.

Table 1: Performance of the outcome classification task on the test set, and on the subset of the test set used in the reader study. We include 95% confidence intervals estimated by 1,000 iterations of the bootstrap method [6]. The optimal weights assigned to the COVID-GMIC and the COVID-GBM predictions in the ensemble were derived by optimizing the AUC on the validation set.

| | Test set (n=832) | | | | | | | |
|---|---|---|---|---|---|---|---|---|
| | AUC | | | | PR AUC | | | |
| | 24 hours | 48 hours | 72 hours | 96 hours | 24 hours | 48 hours | 72 hours | 96 hours |
| COVID-GBM | 0.747 (0.692, 0.796) | 0.739 (0.683, 0.788) | 0.750 (0.701, 0.797) | 0.770 (0.727, 0.813) | 0.230 (0.164, 0.321) | 0.325 (0.254, 0.421) | 0.408 (0.337, 0.499) | **0.523** (0.446, 0.613) |
| COVID-GMIC | 0.695 (0.627, 0.754) | 0.716 (0.661, 0.766) | 0.717 (0.661, 0.766) | 0.738 (0.691, 0.781) | 0.200 (0.140, 0.281) | 0.302 (0.225, 0.395) | 0.374 (0.296, 0.465) | 0.439 (0.363, 0.532) |
| COVID-GBM + COVID-GMIC | **0.765** (0.713, 0.818) | **0.749** (0.700, 0.798) | **0.769** (0.720, 0.814) | **0.786** (0.742, 0.827) | **0.243** (0.187, 0.336) | **0.332** (0.254, 0.427) | **0.439** (0.351, 0.533) | 0.517 (0.434, 0.605) |
| | Reader study dataset (n=200) | | | | | | | |
| | AUC | | | | PR AUC | | | |
| | 24 hours | 48 hours | 72 hours | 96 hours | 24 hours | 48 hours | 72 hours | 96 hours |
| Radiologist A | 0.613 (0.521, 0.707) | 0.645 (0.559, 0.719) | 0.691 (0.612, 0.764) | 0.740 (0.666, 0.806) | 0.346 (0.251, 0.475) | 0.490 (0.381, 0.613) | 0.640 (0.535, 0.744) | 0.742 (0.650, 0.827) |
| Radiologist B | 0.637 (0.544, 0.727) | 0.636 (0.556, 0.720) | 0.658 (0.578, 0.728) | 0.713 (0.640, 0.777) | 0.365 (0.268, 0.501) | 0.460 (0.360, 0.585) | 0.590 (0.479, 0.688) | 0.704 (0.603, 0.792) |
| Radiologist A + Radiologist B | **0.642** (0.555, 0.729) | 0.663 (0.580, 0.737) | 0.692 (0.618, 0.763) | 0.741 (0.673, 0.804) | **0.403** (0.286, 0.534) | 0.499 (0.385, 0.618) | 0.609 (0.507, 0.726) | 0.740 (0.649, 0.830) |
| COVID-GMIC | **0.642** (0.550, 0.730) | **0.701** (0.621, 0.775) | **0.751** (0.681, 0.817) | **0.808** (0.746, 0.866) | 0.381 (0.282, 0.527) | **0.546** (0.435, 0.671) | **0.676** (0.572, 0.788) | **0.789** (0.698, 0.879) |
| COVID-GBM | 0.704 (0.624, 0.776) | 0.719 (0.644, 0.790) | 0.750 (0.679, 0.816) | 0.787 (0.724, 0.847) | 0.411 (0.304, 0.563) | 0.537 (0.434, 0.680) | 0.668 (0.566, 0.778) | 0.804 (0.724, 0.870) |
| COVID-GBM + COVID-GMIC | 0.708 (0.617, 0.779) | 0.702 (0.629, 0.771) | 0.778 (0.705, 0.837) | 0.819 (0.753, 0.875) | 0.411 (0.305, 0.543) | 0.500 (0.399, 0.636) | 0.705 (0.604, 0.811) | 0.808 (0.718, 0.881) |

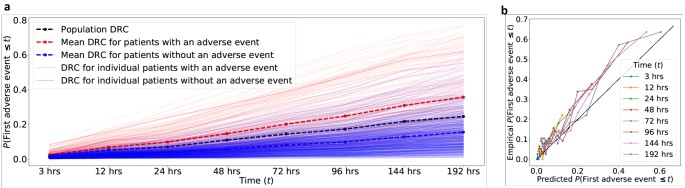

Figure 2: Left to right: the original X-ray image, saliency maps for clinical deterioration within 24, 48, 72, and 96 hours, locations of region-of-interest (ROI) patches, and ROI patches with their associated attention scores.

Figure 3: (Left) The mean DRC for patients with adverse events (red dashed line) is higher than the DRC for patients without adverse events (blue dashed line) at all times. The graph also includes the ground-truth population DRC (black dashed line) computed from the test data. (Right) Reliability plot of the DRCs generated by the COVID-GMIC-DRC model for patients in the test set.

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
