# OpenReview forum: "An artificial intelligence system for predicting the deterioration of COVID-19 patients in the emergency department"
_MIDL.io/2021/Conference/Short — MIDL 2021 Poster_

### Official Review · Reviewer_Je6M · 2021-04-28

**Confidence:** 3
**Final Rating:** 4

**Summary:**

The authors propose a model to estimate probability of deterioration of COVID patients in the emergency departments within 96 hours. The model makes predictions from x-ray images, and these predictions are combined with non-imaging clinical data. Another model is also proposed to estimate the deterioration curve over time.

The paper is well written although very dense. The model architecture is well justified and the results are very promising.

**Strengths:**

* An interesting framework to combine imaging and clinical data to make predictions of disease deterioration in a days time frame.
* Trained and tested in a large dataset
* Attention based explainability provides insight on why the deterioration was estimated.

**Weaknesses:**

* Too little information about the predictions from clinical variables
* No information about statistical significance in the different average values in the result, and particularly between human predictions and the machine.
* No indication of human evaluation of the attention -based extracted patches; or an indication other they do pick up disease-relevant areas, other than for the example shown.
* No discussion of the limitations of the method
* There is no information about a control group -are there any healthy subjects in the study? what would the model predict on a healthy patient?  More information of the severity/ground truth information of the data would be useful.

**Deanonymize Review:**

yes

**Detailed Comments:**


Fig 2: It is unclear what the reader should look for to understand what is changing between 24, 48, 72 and 96 hours. Do the attention scores change with t? How can the attention score for patch 6 be 0.0, what does that mean? It would be interesting to compare those attentions with what a radiographer would focus on, even if just for some examples.

The axes and legends in  figure 3 are way too small.

**Justification Of The Rating:**

The paper is solid, with a well conceived model, a large, diverse dataset, and good results. There are some minor aspects that should be improved and clarified but I think this is overall a solid short paper for MIDL.

**Paper Type:**

both

**Special Issue:**

no

---

### Official Review · Reviewer_xKUs · 2021-05-04

**Confidence:** 5
**Final Rating:** 3

**Summary:**

The paper presents a deep learning-based system for the deterioration of COVID-19 patients into the next 96 hours. The method consists of three different components one component that explores the clinical attributes and two components that are based on the X-rays of the patients. The first of these imaging components provide saliency maps indicating interesting regions of the initial input and the second patient’s risk of deterioration evolves over time in the form of a deterioration risk curve. The authors show that the use of all the components provides better performance than medical experts.


**Strengths:**

- The paper is interesting and easy to follow.
- A deep learning-based framework is proposed that combines imaging with clinical attributes for COVID-19 deterioration.
- Saliency maps, extracted from the original X-rays provide the regions that are more important for the network to take the specific decision, providing additional information about the imaging characteristics.
- An ablation study of the different components of the method together with comparisons with human experts prove the soundness of the method.


**Weaknesses:**

- The paper does not present a comparison with other machine/ deep learning methods in the literature, however, the comparison with the human experts provides a baseline.
- The dataset used is not really presented in the paper, e.g. the type and number of clinical attributes used or the way that the deterioration (t=24,48,72,96) has been defined and in particular what is the first adverse event. These components are important for a better understanding of the method.


**Deanonymize Review:**

no

**Detailed Comments:**

- I think a small discussion about the training and computation complexity of the method is needed in order to prove the soundness of the method.
- How much the number of selected saliency maps affect the results of the method? How the number has been selected?
- The proposed method performs a bit better than the model that is based on only clinical.
- In Table 1 I think that the bold numbers do not really correspond to the best performance of the tested components.
- I think there is a problem with the conference template used for this paper.


**Justification Of The Rating:**

Justification of rating
Overall, I think that this contribution will be interesting for MIDL 2021 both in terms of method and application.
I therefore recommend a weak accept rating for this MIDL paper

**Paper Type:**

methodological development

**Special Issue:**

no

---

### Meta-Review · Program_Chairs · 2021-05-09

**Recommendation:** Accept (Poster)
**Confidence:** 4

**Metareview:**

Both reviewers recommend acceptance. Minor issues as pointed out by the reviewers should be addressed in the final version.

---

### Decision · Program_Chairs · 2021-05-11

Accept (Poster)